# Total Column Ozone Retrieval from a Novel Array Spectroradiometer

Luca Egli[1], Julian Gröbner[1] Herbert Schill[1] and Eliane Maillard Barras[2]

[1]Physikalisch-Meteorologisches Observatorium Davos, World Radiation Center (PMOD/WRC), 7260 Davos Dorf, Switzerland
[2]Federal Office of Meteorology and Climatology, MeteoSwiss, 1530 Payerne, Switzerland

*Correspondence to*: L. Egli (luca.egli@pmodwrc.ch)

**Abstract**

This study presents a new total column ozone (TCO) retrieval from the Koherent system, developed at PMOD/WRC. The instrument is based on a small, cost effective, robust, low-maintenance and state-of-the-art technology array spectroradiometer. It consists of a BTS-2048-UV-S-F array spectroradiometer from Gigahertz-Optik GmbH, coupled with an optical fiber to a lens-based telescope mounted on a sun tracker for measuring direct UV irradiance in the ultraviolet wavelength band between 305 nm to 345 nm.

Two different algorithms are developed for retrieving TCO from these spectral measurements: 1) TCO retrieved by a least squares fit algorithm (LSF) and 2) a Custom Double Ratio (CDR) technique using four specifically selected wavelengths from the spectral measurements. The double ratio technique is analogous to the retrieval algorithm applied for the Dobson and the Brewer but adopted here for TCO retrieval with Koherent. The instrument was calibrated in two different ways: a) absolute calibration of the spectra using the portable reference for ultraviolet radiation QASUME for the LSF retrieval and b) relative calibration of the extraterrestrial constant (ETC) of the CDR retrieval, by minimizing the slope between air mass and the relative differences of TCO from QASUME and Koherent. This adjustment of the ETC allows the instrument to be calibrated with standard TCO reference instruments during calibration campaigns, such as a double monochromator Brewer.

A two-year comparison in Davos, Switzerland, between Koherent and the Brewer 156 (double monochromator) shows that TCO derived from Koherent and the Brewer 156 agree in average over the entire period within 0.7% for all retrievals in terms of offset. The performance in terms of slant path depends on the selected retrieval and the applied corrections. The stray light corrected LSF retrieval exhibits a smaller slant path dependency than the CDR retrieval and performs almost as well as a double monochromator system. The slant path dependency of the CDR is comparable to the slant path dependency of a single Brewer monochromator. The combination of both retrievals leads to performance with an offset close to zero compared to Brewer 156, a seasonal amplitude of the relative difference of 0.08% and a slant path dependency of maximum 1.64%, which is similar to other standard TCO instruments such as single Brewer or Dobson.

Applying the double ratio technique by selecting the wavelengths and slit functions from Brewer and Dobson, respectively, allow the determination of the effective ozone temperature within 3 K on daily averages. With the improved TCO retrieval, Koherent serves as a new low maintenance instrument which could be used to monitor TCO also at remote sites. The TCO retrieval presented here may be applied to other array based spectroradiometers providing direct spectral measurements in the ultraviolet.

## 1 Introduction

The depletion of the stratospheric ozone layer was discovered in the 1970's (e.g. Molina and Rowland 1974, Solomon 1999, Staehlin et al. 2018) and since then has been continuously reported in the Word Meteorological Organisation (WMO) assessment of ozone depletion (e.g. WMO, 2018) on a quadrennial schedule. In order to monitor the evolution of the ozone

layer, accurate instrumentations such as the Dobson spectroradiometer (Dobson, 1968, Komhyr et al., 1989) and the Brewer spectroradiometer (Kerr et al. 1981) have been developed to ensure worldwide long-term observations of total column ozone (TCO) forming a global network. The ozone layer absorbs solar ultraviolet radiation at wavelengths shorter than 350 nm and protects life on Earth's surface from harmful UV radiation. A spectroradiometer allows retrieving TCO by accurately

measuring direct sun irradiance at the Earth's surface in the UV wavelength band and by using an atmospheric model based on the Beer-Lambert Law (Kerr et al., 1988). In the Dobson instruments, prisms are used to select the four specific wavelengths which are used for the ozone retrieval (Evans, 2008). Most of the Dobsons are manually operated and therefore require substantial manpower for the operation at remote sites. The three Swiss Dobsons used in this study have been automated (Stübi et al. 2021). In the 1980's the Brewer spectroradiometer (Kerr et al., 1981, Kerr et al. 1985) was developed as an automatic

device measuring direct solar UV radiation with gratings instead of prisms. The first instruments were single monochromator with one grating to select the wavelengths at the specific slits. Later in the development, Brewers were equipped with two gratings to form double monochromator instruments. Double monochromator devices perform UV measurements with insignificant stray light impact, in contrast to single monochromator Brewers or to Dobsons, which suffer from the effect of stray light at high solar zenith angles and corresponding high air masses. The Brewers were used to form a network of

automated stations, which requires less operational manpower as an advantage for remote sites. However, the network still requires regular maintenance by trained experts.

For the Dobson and the Brewer spectrophotometers, the retrieval of TCO is obtained by the double ratio technique at four specific wavelengths in the UV absorption band (see section 2.2).Comparison studies (e.g. Kerr (2022), Vanicek (2006), Scarnato et al., (2009), Redondas et al. (2014), Gröbner et al. (2021), showed that the greatest consistency between TCO from

Brewer and Dobson can be achieved by using the G14 ozone absorption cross section (Serdyuchenko et al., 2014) and the inclusion of effective ozone temperature from balloon soundings for the Dobson retrieval. Both instruments require the extraterrestrial constant (ETC), which defines the inferred response of the instrument to the irradiance at the top of the atmosphere, as a parameter of the retrieval. This parameter is determined by Langley plot calibration at some specific sites with stable atmospheric conditions (Mauna Loa, Hawaii, US and Izaña, Teneriffe, Spain) or it is determined by on-site

comparison with a regional reference instrument (Köhler et al. 2002, Redondas et al. 2019). In the framework of the Global Atmosphere Watch program (GAW), WMO defined the Brewer and the Dobson instruments as the standard instruments for TCO monitoring from Earth's surface.

In recent years, other instruments measuring direct solar irradiance between 300 nm and 340 nm were developed to retrieve TCO, without the requirement of on-site calibration by reference instruments. In particular array spectroradiometers have the

advantage that a continuous spectral range is measured instead of only discrete wavelength values as by the Brewer or the Dobson. The instruments are small, robust, require low maintenance during their operation and acquire spectra within a time frame of seconds. However, it is also reported that array spectroradiometers suffer from stray light due to the single monochromator setup (Egli et al., 2016).

An example of an array spectroradiometer based system is the Pandora system, which form a global network for atmospheric

composition measurements, including TCO (Herman et al. 2017). The instrument measures solar spectra with an array spectroradiometer and an input optic connected by a fiber to the spectroradiometer. TCO is retrieved by a spectral fitting algorithm wavelength band between 305 nm and 330 nm. Comparison with Dobson instrument shows an averaged difference of 2.1 ± 3.2% (Herman et al. 2017) with the Pandora system. Herman et al. (2017) reported that TCO have been corrected for the impact of effective ozone temperature and for the influence of straylight. Similar to the Pandora, the entrance optics of the

Phaeton system consists of a fiber-coupled telescope connected to an array spectroradiometer (Kouremeti et al., 2008). The spectral measurements from Phaeton are used to retrieve TCO by a DOAS/MAXDOAS technique using the wavelength band between 315 nm and 337 nm. Averaged biases of 0.94% ± 1.26% of TCO from Phaeton are observed by comparison with a

Brewer single monochromator spectroradiometer when using the effective ozone temperature as input for the retrieval (Gertski et al., 2018).

The BiTecSensor (BTS) array spectroradiometer from Gigahertz Optik GmBH (Zuber et al., 2018a) reduces straylight by filtering solar radiation with different bandpass filters inside the array-spectroradiometer (Zuber et al. 2021). Contrary to the aforementioned systems the "BTS Solar" is based on a BTS array spectroradiometer unit with directly mounted diffusor entrance optic and a tube for field of view limitation for irradiance measurements instead of a fiber coupled telescope (Zuber et al., 2018b and Zuber et al 2021). The retrieval algorithm is based on the comparison from two selected wavelength bands

(307 nm to 311 nm  and 319 nm to 324 nm) of the measured data with a lookup table pre-calculated by the libRadtran software package for radiative transfer calculations (Emde et al., 2016). The look -up table is pre-calculated for a specific location and requires ground pressure for correction but does not use the effective ozone temperature. A long-term intercomparison in 2019 and 2020 of the BTS Solar at Hohenpeissenberg, Germany, showed a bias of less than 0.1% with a standard deviation of less than 0.8% compared to Brewer using the Bass and Paur cross section. Stray light effects of BTS are discussed and reported to

be minor for TCO retrieval by the look-up table technique (Zuber et al., 2018a, Zuber et al., 2018b and Zuber et al 2021). Also in Zuber et al., (2021), the Koherent system operated at PMOD/WRC was introduced as an instrument which connects the BTS array spectroradiometer by an optical fiber to a lens-based telescope (see section 2.1). Since Koherent was first built as a research instrument, the configuration with the optical fiber was chosen to test several additional filters with an in-line filter wheel and to test telescopes with different field of views. However, with today's knowledge and for even more simplicity and

robustness, a collimating tube in front of the diffusor, as provided by the BST-Solar (Gigahertz Optics Gmbh), would be a better solution than an optical fiber with telescope. In analogy to Huber et al., (1995) and Egli et al., (2022) a least square (LSF) spectral fitting algorithm (see section 2.2) is applied for TCO retrieval during 2019 and 2020 in Davos, Switzerland. TCO from a first version of the LSF technique exhibit an averaged bias of 1.7% with a standard deviation of 1.4% and a seasonal cycle of about 2% when compared with Brewer 163 using the Bass and Paur cross section (Zuber et. al. 2021). The

first version of the LSF algorithm used in Zuber et al., 2021 did not include the effective ozone temperature as input in the retrieval and it was discussed if the seasonal cycle is caused by remaining straylight effects or to the retrieval at constant effective ozone temperature of 228 K through the seasons. However, it remained unclear which effect caused the seasonal cycle.

Recently, Egli et al., (2022) presented traceable TCO measurements from the world portable reference spectroradiometer for

UV radiation QASUME (Gröbner et al. 2005). QASUME consists of a scanning double monochromator measuring direct solar UV spectral irradiance at wavelength between 305 nm and 345 nm. QASUME is fully characterized in the laboratory with sources that are traceable to primary SI standards with an unbroken calibration procedure and a comprehensive uncertainty analysis for the entire calibration chain (Hülsen et al., 2016 and Gröbner et al., 2017). TCO from these high-quality direct UV spectral measurements are used for an improved, well tested and optimized LSF retrieval algorithm in the wavelength band

between 305 nm and 345 nm.  In Egli et al., (2022) the improved LSF retrieval and the specific setting such as extraterrestrial spectrum, cross-section and aerosol optical depth parametrization, is described in detail and the same algorithm and settings are applied here for the retrieval for traceable TCO measurement from spectral solar measurements of Koherent. A long-term comparison of QASUME TCO with a Brewer double monochromator and a Dobson in Davos, Switzerland showed an offset of about 1% and a seasonal cycle of less than 0.2%. This small seasonal variation indicates the proper inclusion of the effective

ozone temperature for QASUME LSF retrieval and the negligible impact of straylight.

This study aims first to apply the QASUME LSF retrieval algorithm to spectral measurements from the BTS array-spectroradiometer system Koherent including the effective ozone temperature to investigate potential stray light effects of Koherent. Since stray light effects occur from Koherent measurements, a stray light correction will be applied to improve TCO observations from Koherent. The LSF retrieval is a complementary algorithm tested and validated with QASUME, which

allows TCO to be retrieved traceable to SI units, when the system is calibrated on an absolute level. Second, the measured

spectra of Koherent are used as an input to apply a custom double ratio (CDR) technique (in analogy to the Dobson and Brewer retrieval) to further develop the performance of Koherent. The advantage of the CDR technique is that this algorithm allows calibrating the instrument with dedicated ozone reference instruments such as the Brewer in periodic calibration field campaigns or with traceable to SI units instruments such as QASUME, by adjusting the extraterrestrial constant. The study

also investigates the potential of the double ratio technique applied at Koherent to retrieve the effective ozone temperature from ground-based measurements as shown in Kerr et al. (2002).

The results of TCO from Koherent calculated by the new LSF and the CDR technique are compared with the double monochromator Brewer 156 of MeteoSwiss, operated at the World Radiation Center (PMOD/WRC), in Davos, Switzerland. The comparison during the period from 1 October 2019 to 30 September 2021 allows validating the performance of Koherent.

The overall objective is to obtain continuous TCO observations from a small, robust and low maintenance instrument for remote sites with unattended operation.

## 2 Instrument and Retrieval Algorithm

### 2.1 Koherent array spectroradiometer based instrument

The transportable system Koherent was built as a research instrument to test the ability of a novel, array spectroradiometer for

continuous TCO monitoring. As described in Zuber et al. (2021), Koherent is based on a fiber coupled BTS-2048-UV-S-F array spectroradiometer, which is connected to a lens-based imaging telescope (Figure 1). The telescope with a field of view of about +/- 0.6° is mounted on a sun tracker following the sun to measure direct solar irradiance at the measurement platform at PMOD/WRC, Davos, Switzerland at 1560 m a.s.l. (coordinates: 46.81 N, 9.83 E). The small array spectroradiometer is enclosed in a temperature stabilized weather-proofed box to ensure a constant temperature through all seasons of the year. The

BTS is controlled with a python software-routine on an embedded computer inside the box. The raw data is stored immediately on the embedded computer and transferred to an external server on a five minute schedule. During the operation between 2019 and 2021, the system performed reliably in more than 99% of all days and can be considered as technically mature. The time interval of each reading is usually 1 minute since the device needs around 45 seconds for one measurement. However, at conditions with low direct sun irradiance intensity (e.g. cirrus clouds or high solar zenith angles), around 150 seconds are

needed due to increased integration time (Zuber et al. 2021). Since the airmass can change rapidly at high solar zenith angles, higher integration times than 150 sec were not permitted in the measurement program. For example, using 45 seconds integration time, the airmass changes by 0.35% at airmass 3.5 (or SZA of around 75°) and to 1.15 % for integration times of 150 seconds leading to an uncertainty of TCO of the same relative amount.

The postprocessing to prepare the spectra for TCO retrieval was performed off-line with the following steps:

First, converting the raw-counts of the readings to irradiance using in-situ calibration, with the world reference for UV radiation QASUME (Gröbner et al., 2005, Hülsen et al., 2016 Gröbner et al., 2017). Spectral measurements of QASUME are traceable to the primary spectral irradiance standard of the Physikalisch Technische Bundesanstalt (PTB) (Gröbner and Sperfeld, 2005) via secondary standard tungsten halogen 1 kW FEL lamps. The stability of QASUME is monitored in the field with 250 W tungsten halogen lamps on the measurement platform (Egli et al., 2022). The responsivity from Koherent was calibrated by

the spectra from QASUME at local noon using data from four clear sky days between 5 and 15 of September 2020, when the sun irradiance was as stable as possible over time. This is-situ calibration was used for all other days of the two years period. Zuber et al., (2021) highlighted the difficulties of laboratory calibration of the telescope entrance optics with poor reproducibility of the calibration. Therefore, the direct in-situ calibration with the world UV reference QASUME was chosen for most of the results of this study. The field calibration with QASUME showed a standard deviation of 0.96%. Combined

with the uncertainty of QASUME for direct radiation (0.91%, Hülsen et al., 2016) the uncertainty of Koherent spectra is about 1.32%. The lamp calibration will later be used when adjusting the ETC in the double ratio technique (section 3.2 b). Figure 2

displays the ratio between synchronous spectra from QASUME and Koherent for individual wavelength bands normalized to local noon (12 UTC), exhibiting an increase for all wavelength longer than 310 nm of +2% between SZA 60° and 70°. The wavelength band at 305 nm increases after a small decrease during morning, up to 4% at high solar zenith angles. The larger increase at 305 nm is a sign of stray light impact at shorter wavelengths, which will be addressed and discussed later.

The second step of the post processing chain, consisting in a wavelength shift correction and spectrum homogenization using the MatSHIC software developed at PMOD/WRC (Egli et al., 2022), was applied to the Koherent spectrum to obtain a standard spectrum for further retrieval of TCO. For this study, the resulting spectra were homogenized to 0.01 nm wavelength-increment and convolved with a triangular slit function of 0.5 nm full width half maximum for the least square (LSF) fit algorithm (see section 2.2) and to 0.1 nm for the custom double ratio technique (section 2.3), respectively. The so homogenized spectra were used to retrieve TCO with the following algorithms.

## 2.2 Least Square (LSF) fitting algorithm

Egli et al. (2022) presented a revised algorithm to retrieve TCO from spectral irradiance measurements. The same algorithm, thoroughly tested in Egli et al. (2022), is applied here for TCO retrieval from Koherent. The algorithm used here differs from the algorithm used in Zuber et al. (2021) by the inclusion of the effective ozone temperature as input parameter and differs in other specific settings (e.g. solar spectrum TSIS, Coddington et al., 2021) as it was shown to be the best configuration in Egli et al. (2022).

In summary, the LSF algorithm is using a spectral least square fitting procedure by minimizing the residuals of the fit in the wavelength range between 305 nm to 345 nm and implements an atmospheric model based on the Beer-Lambert law:

$$I_\lambda(T, p, O3) = I_\lambda^0 exp[-(\tau_\lambda^{O3}(T) \cdot m^{O3} + \tau_\lambda^{AOD} \cdot m^{AOD} + \tau_\lambda^R(p) \cdot m^R + \tau_\lambda^{SO2} \cdot m^{SO2})] \qquad \text{Eq.1}$$

$I_\lambda$ denotes the modelled spectral irradiance at the wavelength $\lambda$. $I_\lambda^0$ indicates the TSIS solar reference spectrum at the top of the atmosphere (Coddington et al., 2021), while the airmass $m$, denotes the path length of radiation through the atmosphere. The atmospheric model accounts for the effect of ozone absorption at each wavelength and furthermore the attenuation by the atmosphere including aerosols, Rayleigh scattering and Sulfur Dioxide (SO₂). The resulting attenuated modelled solar spectrum using Eq. 1 is then compared with the measured solar spectrum by Koherent, by minimizing the spectral residuals between the modelled and measured solar spectrum and returns the corresponding model parameters including TCO (Egli et al., 2022).

For the term of ozone attenuation $\tau_\lambda^{O3}(T) \cdot m^{O3}$ in Eq. 1, the SG14 ozone absorption cross section (Serduychenko et al. 2014, Egli et al. 2022) for different effective ozone temperatures ($T$) is used in the retrieval algorithm. TCO retrieved with SG14 shows a dependency on effective ozone temperature of 0.1%/K (Egli et al. 2022). This implies, that the effective ozone temperature is required as input for the TCO retrieval algorithm to achieve uncertainties in retrieved TCO of less than about 1%. In analogy to Gröbner et al. (2021) and Egli et al. (2022), the effective ozone temperature retrieved from balloon sounding at the nearest sounding station in Payerne, Switzerland, is taken as daily input for the retrieval algorithm. Since ozone soundings take place three times a week, the effective temperature from ozone sondes were linearly interpolated to daily values. Moreover, ozone and temperature above the sonde burst height are calculated by extending the measured ozone and temperature profiles with the standard ozone and temperatures taken from the US Standard Atmosphere temperature and ozone density values just below the sonde burst height, as in Gröbner et al. (2021). The term $\tau_\lambda^{AOD} \cdot m^{AOD}$ in Eq. 1 denotes the

attenuation by aerosol optical depth. The wavelength dependence of the aerosol optical depth is parametrized linearly, normalized to 340 nm. The term $\tau_\lambda^R(p) \cdot m^R$ of Eq. 1 accounts for the effect of Rayleigh scattering in the atmosphere, which is parametrized according to Bodhaine et al. (1999) and requires the air surface pressure $p$. Since the LSF retrieval is insensitive to pressure changes of +/- 7 hPa in Davos, which results in +/- 0.014% in TCO (Egli et al. 2022) $p$ is selected from the pressure climatology in Davos as 820 hPa for Davos. The attenuation by $SO_2$ is also considered in the overall attenuation equation (Eq.1), even though the amount of $SO_2$ in the atmosphere above Davos is insignificant (Gröbner et al., 2021). Accounting for all four terms of attenuation in the Beer-Lambert atmospheric model, the LSF approach derives the best fit to the unknown parameters AOD and TCO. The airmass $m$, denoting the path length of radiation through the atmosphere is calculated based on the geometry between the Earth, atmosphere and the sun for each time stamp when the spectrum was measured. The airmass for the ozone ($m^{O3}$), aerosol ($m^{AOD}$) and Rayleigh ($m^R$) and $SO_2$ ($m_\lambda^{SO2}$) is calculated from the standard US atmosphere profile for mid-latitudes (NOAA, 1976) as applied in Egli et al. (2022).

**2.3 Custom Double Ratio (CDR) Technique**

A second approach to retrieve TCO from the spectral measurements is to apply the double ratio technique from four specific wavelength as used for the Brewer TCO retrieval (Kerr et al., 1985). This allows Koherent to be calibrated by adjusting the extraterrestrial constant in the same manner as other instruments using the double ratio technique. In contrast, the LSF retrieval requires calibrated spectra from a spectral reference such as QASUME. Spectral measurements from Koherent allow to select specific wavelengths from the measured spectrum and to apply the double ratio technique as for the Brewer.

Following the notation of Gröbner et al. (2021), who summarized the double ratio technique for Brewer and Dobson, we present here the principle of the custom double ratio (CDR) technique for Koherent as follows:

In analogy to the LSF retrieval, the absorption through the atmosphere is based on the Beer-Lambert Law (Eq. 1). The equation can be rewritten for the CDR technique as:

$$\tau_\lambda = \frac{\log I_0^{CDR}(\lambda) - \log I^{CDR}(\lambda)}{m} \qquad \text{Eq. 2}$$

Where $I$ is the measured spectrum, $I_0$ is the top of atmosphere reference spectrum and $\tau_\lambda$ is the total optical depth from the different absorption and attenuation terms. Contrary to Eq. 1 where $\lambda$ indicates each individual wavelength of the measured spectrum, $\lambda$ denotes one of the four wavelengths for the double ratio technique. For the CDR we have selected the four wavelengths: $\lambda_i^{CDR} = [310, 322, 332, 345]\ nm$ to obtain a significant sensitivity to ozone absorption. The selection of the wavelength is empirically based on a shift of 5 nm from the nominal wavelengths of the Dobson ($\lambda_i^{Dobson\ 101} = [305.4, 324.9, 317.4, 339.7]$). These wavelengths revealed a better performance in terms of offset and seasonal amplitude than other wavelengths within these wavelength ranges. However, all possible combinations of wavelength setting were not tested. Moreover, $I^{CDR}(\lambda)$ is not the solar irradiance at only one specific wavelength of the measured spectrum, it indicates the integral of the spectrum over defined slit functions $s_i(\lambda)$ (Figure 3 and Eq. 3) with the center-wavelength at $\lambda_i^{CDR}$.

$$I_i^{CDR}(\lambda) = \frac{\int I_i(\lambda) \cdot s_i(\lambda)\, d\lambda}{\int s_i(\lambda)\, d\lambda} \qquad \text{Eq. 3}$$

The index *CDR* indicates the integral over the slit function $s_i(\lambda)$ (Eq. 3), not only of the measured spectrum, but later also of the solar spectrum, the ozone absorption cross section and the Rayleigh scattering. The slit functions $s_i(\lambda)$ for the CDR are rectangular slit functions with the width of 1 nm for the wavelength 310 nm and 322 nm (i=1,2) and the width of 4 nm for the wavelengths 330 and 345 nm (i=3,4), respectively (see Figure 3, blue rectangles). The different sizes of the widths for shorter

and longer wavelengths are chosen in analogy to the Dobson slits, where the width at shorter wavelengths is about 1 nm and about 4 nm at longer wavelengths.

As stated in Gröbner et al. (2021), the effect of Sulphur dioxide and nitrogen dioxide can be neglected for the atmosphere in Davos. However, the attenuation from Rayleigh scattering $\tau_\lambda^R(p) \cdot m_\lambda^R$ needs to be subtracted from the total optical depth $\tau$ to retrieve the optical depth for ozone in the UV band:

$$\tau_{O3}(\lambda, T, p) = \alpha(\lambda, T) \cdot TCO \cdot m^{O3} = \frac{\log I_0^{CDR}(\lambda) - \log I^{CDR}(\lambda) - \tau_\lambda^R(p) \cdot m^R}{m^{O3}} \quad \text{Eq. 4}$$

Where $m^{O3}$ and $m^R$ are the effective airmass of ozone and Rayleigh-scattering. Gröbner et al. (2021) showed with balloon soundings from Payerne that the effective ozone height can change from 20.2 km to 23.8 km (mean = 22.3 km) within a season. For comparability with Brewer 156 we have also chosen here a constant ozone layer of 22 km and the effective molecular scattering height relevant for Rayleigh scattering of 5 km, which is the nominal operating procedure for Brewers. The difference between the effective ozone height and 22 km results in a maximum uncertainty of 0.3% of TCO at the highest air mass of 3.9 in Davos. This effect is considered in the overall uncertainty budget of the CDR algorithm (section 3.3). $\alpha(\lambda, T)$ is the SG14 cross section depending on effective ozone temperature as described later. The influence of absorption and scattering by aerosols in the atmosphere is minimized by using ratios of measurements at close wavelengths. TCO is then retrieved by a linear combination of the measurements, the solar spectrum and the Rayleigh scattering term from the four specific wavelengths $\lambda_i^{CDR}$,

$$TCO = \frac{F^{CDR} - F_0^{CDR} - \Delta\beta \cdot m^R}{\Delta\alpha \cdot m^{O3}} \quad \text{Eq. 5}$$

where $F^{CDR} = \sum_{i=1}^{4} w_i \cdot \log I^{CDR}(\lambda_i^{CDR})$ is the logarithmic sum of the solar irradiance. Accordingly, the terms for the top of the atmosphere solar irradiance $F_0^{CDR}$ is calculated by the weighted sum of logarithmic solar reference spectrum TSIS (Coddington et al., 2021). For the CDR algorithm used here, the weights are equally chosen in analogy to the Dobson retrieval $w_i^{CDR} = [+1 \ -1 \ -1 \ +1]$ to calculate the ratios of measurements for best performance. Finally, since the absorption cross section are measured in cm$^{-2}$ TCO in Eq. 5 was multiplied by the factor 1000 to obtain DU.

The ozone absorption coefficient $\alpha_i$ at wavelength $\lambda_i$ is derived by the SG14 (Serdruchenko et al., 2014) cross section $SG14(\lambda, T)$, where $T$ is the effective ozone temperature from balloon soundings in Payerne, Switzerland (see section 2.2). The ozone absorption coefficient is calculated for the four wavelengths by the same convolution as described in Eq. 3, replacing $I_i(\lambda)$ by $SG14(\lambda, T)$.

The weighted ozone absorption coefficient $\Delta\alpha$ for Eq. 5 is then derived by the weighted sum of $\alpha_i$: $\Delta\alpha = \sum_{i=1}^{4} w_i \cdot \alpha_i$.

The Rayleigh attenuation term $\Delta\beta$ for Eq. 5 is calculated again based on Eq. 3 and the weighted sum by replacing $I_i(\lambda)$ by the parametrization according to Bodhaine et al. (1999), requiring the climatological air surface pressure $p = 820$ hPa in Davos, Switzerland, as input. A sensitivity analysis showed that TCO is 0.09% sensitive on pressure changes of +/- 7 hPa in Davos.

The algorithm presented above can also be adapted to apply a double ratio Brewer retrieval with Koherent spectra, when using the trapezoidal slit functions of the Brewer with full width half maximum (FWHM) of around 0.55 nm at wavelengths $\lambda_i^{Brewer} = [310.0, 313.5, 316.8, 320.1]$ nm (Figure 3, green lines) with weightings of $w_i^{Brewer} = [+1 \ -0.5 \ -2.2 \ +1.7]$ (Gröbner et al. 2021).

Analogously, the Dobson retrieval can be applied by using the measured slit functions of the automated Dobson D101 (Stübi et al. 2021) derived from a tuneable and portable radiation source (TuPS, Smid et al., (2020)). The measurement of the slit functions revealed center-wavelengths of $\lambda_i^{Dobson\ 101} = [305.4, 324.9, 317.4, 339.7]$ and FWHM of around 1 nm for the two shorter wavelengths and around 4 nm for the two longer wavelengths (Köhler et al., 2018). The slit functions are shown in

Figure 3 for the Brewers (trapezoidal slit functions), the Dobsons (measured by TuPS) and the rectangular shapes for the CDR technique introduced here. The slit functions, center-wavelengths and the weightings of the CDR retrieval differ from the Brewer or Dobson wavelengths and are chosen for Koherent for best performance of the CDR retrieval with Koherent. The performance will be presented and discussed in the following section.

## 3 Results and Discussion

The results of TCO from Koherent are compared with TCO from the Brewer 156 double monochromator operated at PMOD/WRC. Due to the long-term quality assurance and strong straylight suppression of the Brewer 156, the instrument was chosen to serve as a reference for the two-year period between 1 October 2019 and 30 September 2021. Additionally, some aspects of Koherent are compared with two single Brewers 040 and 072 and Dobson 101 from MeteoSwiss operated at PMOD/WRC. Koherent, Brewer 072 and the Dobson 101 were co-located in Davos, Switzerland, during the entire period of comparison. Brewer 040 moved to Davos in February 2021. In contrast to Zuber et al. (2021), who used the standard Brewer TCO with Bass and Paur cross section (Bass and Paur, 1985), TCO from the Brewer 156 instrument is calculated with the SG14 cross-section as presented in Gröbner et al. (2021). The following graphs show the relative differences of quasi simultaneous TCO within a five-minute interval in relation to TCO from the Brewer 156 indicated in percent. The missing "period" of TCO comparisons in the figures are attributed to some testing of Koherent and have been removed from the comparison.

In the following, the comparison is quantified in terms of offset, which indicates the averaged differences between the instruments, seasonal amplitude, which shows the amount of seasonal effect by displaying a sinusoidal fit to the differences The slant path dependency is the relation between the relative differences and the ozone slant path column, which is defined as the product of TCO and airmass. To quantify the ozone slant path dependency, the difference of maximum and the minimum of the quadratic fit in the slant path column between 300 DU and 1200 DU is calculated.

### 3.1 LSF retrieval

*a) Wavelength range 305 nm to 345 nm*

Table 1 lists the comparison of TCO between Brewer 156 and Koherent when applying the LSF retrieval and using the spectral range between 305 nm and 345 nm for the spectral fitting. The results show that in average over the entire period, Koherent slightly overestimates TCO with an offset of 0.26%. However, a significant seasonal amplitude of 1.17% is observed as well. Zuber et al. (2021) compared Koherent to the double monochromator Brewer 163 using a first version of the LSF algorithm without taking into account the effective ozone temperature as input parameter and using the Bass and Paur cross-section (Bass and Paur, 1985). Their results show a standard deviation around 2.72 % with an offset of -1.64% and a seasonal amplitude of around 2%. Our result indicates that the inclusion of the effective ozone temperature improves the agreement of Koherent and Brewer double monochromator with SG14 cross section.

Figure 4 displays the dependency of the relative differences on the ozone slant path column. The circles are averages of the five minute values between intervals of 100 DU of ozone slant path column. The lines are a quadratic fit of the five minute values, which are not shown in the Figure. The graph shows a strong dependency of the relative TCO differences with ozone slant column of 4.42% (Table 1). This figure and the seasonal cycle indicate that the biases are larger at higher air masses or high solar zenith angles. The observed seasonal cycle may be caused by straylight at shorter wavelengths. This hypothesis is supported by the results in Figure 2 (lower panel), where solar spectra from Koherent deviate more at shorter wavelengths (305 nm) than at wavelengths longer than 310 nm with respect to the QASUME solar reference spectra. In order to further support this hypothesis, the wavelength range of the LSF retrieval was changed to 310 nm to 345 nm.

 *b) Wavelength range 310 nm to 345 nm*

Restricting the wavelength range from 310 nm to 345 nm, the seasonal cycle can be reduced to 0.86% with an offset close to zero (Table 1). Figure 4 also shows a lower ozone slant path dependency of 2.18% between maximum and minimum than for the 305 nm to 345 nm wavelength range. As shown in Figure 2 at low solar zenith angles (SZA), all wavelength regions are within 0.5%. At higher SZA (> 60°), the wavelength region around 305 nm deviates up to 2% (at SZA 70°) compared to the other wavelengths. This spectral bias explains the slant path dependency change when restricting the wavelength range to 310 nm to 345 nm.

As shown in section 2.1, the measuring-time for each spectrum can vary from 45 seconds up to 150 seconds. The longer measurement-time of 150 seconds at airmass of 3.5 leads to airmass changes up to 1.15 % during this time span. This effect may also contribute to the observed slant path dependency but is minor compared to the contribution of stray light.

*c) Stray light correction*

An alternative method to account for the effect of stray light is to apply a correction on the TCO retrieved by LSF. For this purpose, four full clear sky days between 5 and 15 September 2022 were chosen to parametrize the slant path dependency between the LSF retrieved TCO of QASUME (Egli et al. 2022) and the LSF retrieved TCO of Koherent by a linear fit. Both LSF retrievals used the wavelength range between 305 nm and 345 nm. The coefficients of this linear fit were then applied to all LSF retrievals from Koherent for the entire two years period. Figure 4 (red line) shows a resulting improved slant path dependency of 0.46%. For comparison, the slant path dependency between the two double monochromator instruments Brewer 156 and QASUME is displayed in Figure 4 (grey line). The maximal difference between the 2 instruments shown by the grey line in Figure 4 is 0.18% (Table 1) which is similar to the slant path dependency between two Brewer double monochromator of 0.35% (Brewer 163 vs. Brewer 156, Table 1) and is in the range of the stray light corrected LSF retrieval. However, the red circles in Figure 4 show that the slant path dependency of stray light corrected Koherent LSF is similar to the QASUME-Brewer 156 comparison restricted to ozone slant path between 300 and 900 DU. For ozone slant path higher than 900 DU, Koherent shows a significant bias, which is not observed between the two double monochromator Brewers.

## 3.2 Custom double ratio technique

*a) CDR from calibrated spectra*

As described in section 2.3, TCO can also be retrieved by the CDR technique using the four specific wavelengths and the rectangular slit functions (Figure 3). Table 1 presents the results of the two-year comparison of Koherent CDR with the Brewer 156 double monochromator. The amplitude of the seasonal cycle is about 0.17% with an offset of 0.02%. This low amplitude is substantially reduced compared to the LSF retrieval. Figure 5 blue circles and line shows the relative differences between TCO of Brewer 156 and Koherent dependent on the ozone slant path. The minimum to maximum dependency of the quadratic fit on the slant path is around 2.1% (Table 1).

For comparison, Figure 5 gray circles and line shows the slant path dependency of the relative differences between Brewer 156 and the single monochromator Brewer 072. The maximum to minimum of the quadratic fit of the data reveals a slant path dependency of 2.53% (Table 1). Table 1 also lists the slant path dependency of single monochromator Brewer 040 with Brewer 156 which is 1.52%. The performance of Koherent CDR lies between the two single Brewers 040 and 072, which indicates that the TCO retrieval from Koherent CDR performs at least as well as from a single Brewer monochromator, with respect to ozone slant path. This CDR retrieval algorithm does not require any field calibration of TCO by adjusting the extraterrestrial constant (ETC) or the ozone absorption coefficient with a reference instrument as it is required for the standard double ratio technique with Brewer or Dobson. Instead, this CDR algorithm requires only a one-time vicarious calibration of the spectra

with QASUME, the laboratory measured ozone cross sections (IUP, Serdyuschenko et al. 2014), Rayleigh scattering absorption coefficients (Bodhaine et al., 1999) and the space borne top-of-atmosphere solar reference spectra (Coddington et al. 2021) for the extraterrestrial constant ( $F_0^{CDR}$ , Eq. 5).

380

### b) CDR with adjusted ETC

Since CDR is analogous to the Brewer and Dobson retrieval algorithm, which requires the determination of the extraterrestrial constant by comparison with a reference instrument (Redondas et al. 2019, Komhyr 1989), $F_0^{CDR}$ (Eq.5) could also be determined by an intercomparison with a reference instrument. For Dobsons, only the ETC is adjusted according to the reference Dobson, since the ozone absorption coefficient is set to a nominal value (Komhyr, 1989), assuming that the slits have been correctly adjusted during calibration of the Dobson.. For Brewers, the adjustment of ETC is provided by a dedicated reference Brewer e.g. from the Regional Brewer Calibration Center Europe (RBCC-E) by periodic in-situ intercomparison campaigns (Redondas et al., 2019). Since the reference Brewer is calibrated on an absolute scale and the ozone absorption coefficient may vary specifically for each instrument, not only the ETC, but also the absorption coefficient $\Delta\alpha$ (Eq. 7) can be adjusted in a so called two-point-calibration, where both the ETC and the ozone absorption coefficient are retrieved from the intercomparison with a reference Brewer.

In order to avoid the calibration and validation with the same instrument, we have chosen an alternative approach to determine the ETC for Koherent by using QASUME as a reference and validating the resulting TCO with Brewer 156. Since changes of the ETC affect the ozone slant path dependency, the adjustment of Koherent's ETC is performed by minimizing the slope of the dependency between air mass and the relative difference of QASUME and Koherent. Using the laboratory-based calibration of Koherent instead of the QASUME in-situ calibration, four clear sky days between 5 and 15 September 2020 have been chosen to adjust the extraterrestrial constant $F_0^{CDR}$ of Koherent in comparison with QASUME. This adjustment displays a relative calibration using the relative differences of QASUME and Koherent only. To avoid the inclusion of absolute TCO of this calibration we have further chosen the air mass only as a regression parameter instead of the slant path. Finally, to further minimize the effect of stray light from Koherent, an air mass range less than 2 for minimizing the slope is used.

The so adjusted ETC was used in the CDR algorithm to test the performance of the procedure. The two-year comparison to Brewer 156 of Koherent TCO retrieved by CDR with the adjusted ETC shows an offset of -0.59% and a seasonal amplitude of 0.34% (Table 1) and is therefore well within the performance of the other retrievals (Table 1) or within comparisons of Arosa/Davos Brewers (Stübi et al. 2017 a,b). Figure 5 green circles and green line show that the slant path dependency of around 2% from maximum to minimum, which is similar as for the CDR retrieval and which is also less than the performance of Brewer 072 (Figure 5, grey circles and line). The adjustment of the ETC by minimizing the slope of the air mass dependency is not a standard method but is used here as an alternative method for determining the ETC with QASUME without any QASUME spectral or absolute TCO calibration. However, as aforementioned, the CDR algorithm can be calibrated with well-established calibration procedure if a network reference instrument such as the Brewer from RBCC-E is available (Redondas et al., 2019). This study shows that some calibration days during a field campaign are sufficient to obtain an ETC calibration which is stable over the two years.

**3.3 Combining LSF and CDR**

Since the methods presented in section 2 are different, but scientifically valid approaches, and since the validation with Brewer 156 showed comparable results, the three methods (LSF, CDR and ETC adjusted CDR) were merged to obtain a combined dataset of TCO from Koherent, by averaging all TCO values from the three methods. This averaged product displays a general

estimation of TCO from the different methods, while the individual retrieval can be used for different applications for observing TCO. In order to obtain such a general TCO estimation, we have merged the stray light corrected LSF, the CDR and the ETC adjusted CDR dataset to one combined dataset. Figure 6 shows the entire temporal course between 1 Oct 2019 and 30 September 2021 of the combined TCO dataset. The small blue points indicate the five minutes measurements and the light blue circles monthly averages. The offset between combined TCO and Brewer 156 is close to zero with a seasonal amplitude

of 0.08%. Figure 5 red circles and line and Table 1 additionally highlights that the slant path dependency is comparable to the performance of a single monochromator Brewer. The averaging of the three different retrievals results allows determining the standard deviation of each retrieval. The grey area in Figure 6 indicates the mean relative standard deviation of the individual retrieval to their average. The resulting 0.88% can be considered as the lower level of uncertainty (k=1) of TCO retrieved by Koherent when using different methods and calibration procedures.

In analogy to Egli et al. (2022) for QASUME, a Monte Carlo uncertainty budget for the Koherent LSF retrieval was calculated by variation of all input uncertainties of the parameters described in Eq. 1. The measurement uncertainty of Koherent is 1.3% (see section 2.1). The contribution of the other uncertainties remains the same as in Egli et al. (2022), since the same algorithm with the same input parameters is used for Koherent as for QASUME. The overall uncertainty budget for Koherent derived from Monte Carlo simulations results in an uncertainty of 0.95%.

For the CDR algorithm, the overall uncertainty budget is also derived by a Monte Carlo simulation. The measurement uncertainty (1.3%, section 2.1), the uncertainty from ozone absorption cross section (1.5%, Serduychenko et al. 2014), extraterrestrial spectrum (1.3%, Coddington et al., 2021), ground pressure (7hPa, Egli et al. 2022), ozone air mass (0.086% = $0.3\%/2/\sqrt{3}$ et al. 2022), wavelength shift (0.05 nm from MatSHIC preprocessing) and effective ozone temperature (2.5 K, Egli et al. 2022) was varied as gaussian noise in multiple ensemble runs (500). The standard deviation of these runs revealed

an overall uncertainty budget of 0.64%. One may argue that gaussian noise applied on the entire spectrum may not be an appropriate assumption e.g. for the absorption cross section, the measurement and the extraterrestrial spectrum. An additional Monte Carlo simulation, where the entire spectra of cross section, extraterrestrial spectrum end measurement spectrum were shifted around the input uncertainty, showed an overall uncertainty of 1.57%. Therefore, we estimate the overall uncertainty from the CDR algorithm to be about 1.6%

As described in section 2, the effective ozone temperature is required for all TCO retrievals from Koherent measurements. The dependency on effective ozone temperature of the LSF retrieval is around 0.1%/K, which is in line with the temperature dependency of traceable TCO measurements with QASUME (Egli et al., 2022) or the effective temperature dependency of Dobsons TCO (Redondas et al., 2014). The CDR technique exhibits a slightly larger effective ozone temperature dependency of 0.16%/K. In order to account for this temperature dependency, $\Delta\alpha = 0.7513$ in Eq. 5, needs to be corrected by applying

0.16%/K of the effective ozone temperature measured by balloon soundings (Gröbner et al., 2021) or from ECMWF reanalysis data (https://www.temis.nl/climate/efftemp/overpass.php, last access: 23 November 2022). However, as will be described in the following section, the effective ozone temperature can be directly retrieved by Koherent.

### 3.4 Correlation with effective ozone temperature


Kerr (2002) reported that the specific wavelength settings and the weighting for the double ratio technique of the Brewer retrieval led to a low sensitivity of TCO to effective ozone temperature. Kerr (2002) further demonstrated that the effective ozone temperature can be retrieved from Brewer data with the group-scan method analysing different wavelength settings of the Brewer. Redondas et al. (2014) also showed a sensitivity on effective ozone temperature of about 0.1%/K for the Dobson

retrieval. In other words, if TCO is retrieved with a constant effective temperature for Brewer and Dobson, a seasonal cycle depending on the seasonal variation of the effective ozone temperature is observed (Redondas et al., 2014). As mentioned in

section 2.3 the CDR technique can also be applied using the different Dobson and the Brewer settings in terms of specific wavelengths, slit functions and weightings (Gröbner et al., 2021).

Here, we estimate the differences in TCO by applying the double ratio technique using the Brewer and the Dobson different wavelength and slit function settings on Koherent spectra for a constant effective ozone temperature of 228 K, in order to investigate the potential of retrieving effective ozone temperature. The relative difference between TCO from the Brewer and the Dobson settings show a strong seasonal cycle of 3.9% amplitude (not shown). The amplitude of the differences between Brewer and Dobson settings used in the CDR retrieval correlates with the effective ozone temperature as shown in Figure 7. Since effective ozone temperature is commonly given on a daily schedule (e.g. ECMWF reanalysis data or interpolated from balloon soundings Payerne), the differences in effective ozone temperatures derived from the Brewer and Dobson settings in the CDR retrieval are averaged to daily values to reduce the variability of the individual five minute values. Figure 7 displays a linear fit to the data. The linear fit gives a sensitivity of the TCO on effective ozone temperature of 0.52%/K. Based on the linear parametrization, the effective ozone temperature can be reconstructed from the differences of the TCO retrieved from Koherent spectra applying the Brewer and Dobson settings for the double ratio technique. As shown in Figure 8, the so determined effective ozone temperature agrees with the balloon sounding within a standard deviation of circa 3 K. For comparison, the effective ozone temperature from ECMWF reanalysis data generally agrees with balloon sounding within 2.5 K (Gröbner et al. 2021). Estimates of the effective ozone temperature with Koherent show a larger day-to-day variation than balloon soundings measurements or ECMWF reanalysis. However, when averaging the temperature derived from Koherent on a weekly basis, the agreement is within 2.5 K.

For comparison, Kerr (2002) achieved a typical standard deviation of a daily set of measurements of 0.8 K by the group scan method with Brewers, in Toronto, Canada and Mauna Loa, Hawaii. Differently to Kerr (2020) the retrieval of effective ozone temperature with Koherent is based on a parametrization with data from the two-year intercomparison in Davos, Switzerland. It is unclear if the same parametrization can be applied for other stations worldwide, but it can at least be applied to determine the effective ozone temperature in Davos, within 3 K on a daily basis. This has the significant advantage that the TCO retrieval of Koherent is self-sufficient.

## 4    Conclusion and Summary

In addition to the results already published in Zuber et al. (2021), we have shown that TCO can be retrieved with an improved least square fit retrieval and a new custom double ratio retrieval which allows calibration the instrument with standard reference instruments in filed campaigns. Both retrievals are based on direct solar ultraviolet irradiance spectra measured by an array spectroradiometer based system (Koherent), which is a small and cost-effective instrument requiring very little maintenance. The instrument was operated during two years with more than 99% data acquisition reliability. We have investigated three different retrieval methods, namely the linear least square fit (LSF), stray light corrected LSF, Custom Double Ratio technique (CDR) and CDR with adjusted extraterrestrial constant. The straylight corrected LSF retrieval shows a slant path dependency, which is comparable to a double monochromator but shows a seasonal amplitude of 0.30% (Table 1). In contrast, the seasonal amplitude of CDR is around 0.17%, but with a slant path dependency of 2.1%, which is comparable to the slant path dependency of a single Brewer (Table 1). The overall uncertainty of the LSF retrieval is about 1%, while uncertainty budget of the CDR retrieval is about 1.6%.

We have shown that TCO from Koherent can either be retrieved by an absolute calibration of the Koherent spectra with the portable world reference for ultraviolet radiation QASUME, or by an adjustment of the ETC. The calibration of the ETC allows the instrument to be calibrated with reference instruments during intercomparison campaigns. Furthermore, absolute TCO can be calibrated with a two-point calibration during such campaigns by additionally changing the absorption coefficient to achieve the best agreement to existing network instruments (Redondas et al., 2019).

The double ratio technique using the Brewer and Dobson wavelength settings applied on Koherent spectra, further allows to determine the effective ozone temperature within 3 K as a daily average. The retrieval of effective ozone temperature requires a parametrization of Koherent data with an external effective ozone temperature dataset. Once the parametrization with effective ozone temperature is provided with e.g. data of one year, the parametrization leads Koherent to serve as a standalone instrument for TCO observation without the need of external supporting data such as e.g. daily effective ozone temperature

obtained from balloon soundings or other instruments.

The retrieval methods from direct ultraviolet spectra presented here are in good agreement with the performance of other TCO instruments, e.g. Brewer or Dobson. The methods presented in this study may be applied to other array spectroradiometer systems measuring direct solar irradiance (e.g. Pandora or BTS Solar) at other stations. Koherent serves now as an instrument for operational TCO monitoring at PMOD/WRC in addition to the existing TCO instrument park located in Davos.


*Competing interests.* The authors declare that they have no conflict of interest.

*Acknowledgement:* This research has been supported by the ESA project QA4EO, grant no. QA4EO/SER/SUB/09 and by GAW-CH MeteoSwiss (project INFO3RS, grant no. 123001926). We further thank two anonymous reviewers for their helpful

comments substantially improving this publication.

*Author contributions:* LE and JG developed the Koherent retrieval algorithm, analysed the data and have written the manuscript. HS and EMB were responsible for the Brewer and Dobson measurements used in this study and revised the manuscript.


*Data availability:* The data are available from the main author (LE) on request.

*Code availability:* The TCO retrieval algorithms are available from the main author (LE) on request.

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

## Comparison with Brewer 156

| Instrument | Longterm Offset [%] | Seasonal Amplitude [%] | Slant Path (Min-Max) [%] |
|---|---|---|---|
| | | | |
| Koherent: LSF 305 nm - 345 nm | 0.26 | 1.17 | 4.42 |
| Koherent: LSF 310 nm - 345 nm | 0.02 | 0.86 | 2.18 |
| Koherent: LSF Stray Light Corrected | 0.67 | 0.30 | 0.46 |
| Koherent: CDR | 0.02 | 0.17 | 2.13 |
| Koherent: CDR ETC Adjusted | -0.59 | 0.34 | 2.12 |
| Koherent: LSF/CDR Combined | 0.05 | 0.08 | 1.62 |
| Brewer 040 | -0.29 | 0.16 | 1.52 |
| Brewer 072 | -0.15 | 0.25 | 2.53 |
| Brewer 163 | 0.21 | 0.02 | 0.35 |
| Dobson 101 | 0.06 | 0.17 | 1.42 |
| QASUME | 1.13 | 0.22 | 0.18 |


*Table 1: Comparison of the performance of different TCO retrievals for Koherent and other TCO instruments. The double monochromator Brewer 156 is chosen as the reference instrument for the comparison. LSF stands for Least Square Fit retrieval and CDR for Custom Double Ratio technique. The offsets indicate the long-term bias over the entire period of comparison (1 oct 2019 to 30 Sep 2021). The seasonal amplitude indicates the amplitude of a sinusoidal fit of the TCO relative differences over the years (see Figure 6) and the slant*
*path quantification indicates the difference of the ozone slant path dependency of a quadratic fit of the TCO relative differences (see Figures 3 and 4).*

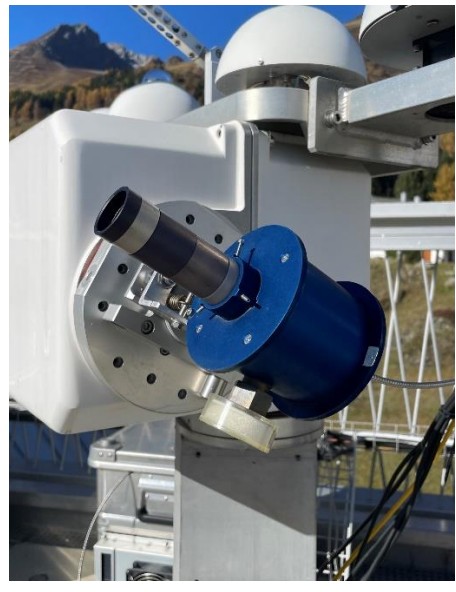


*Figure 1: The lens-based telescope of Koherent on a sun tracker at the measurement platform of PMOD/WRC. The telescope is connected by an optical fiber to the BTS array spectroradiometer in the temperature stabilized box.*

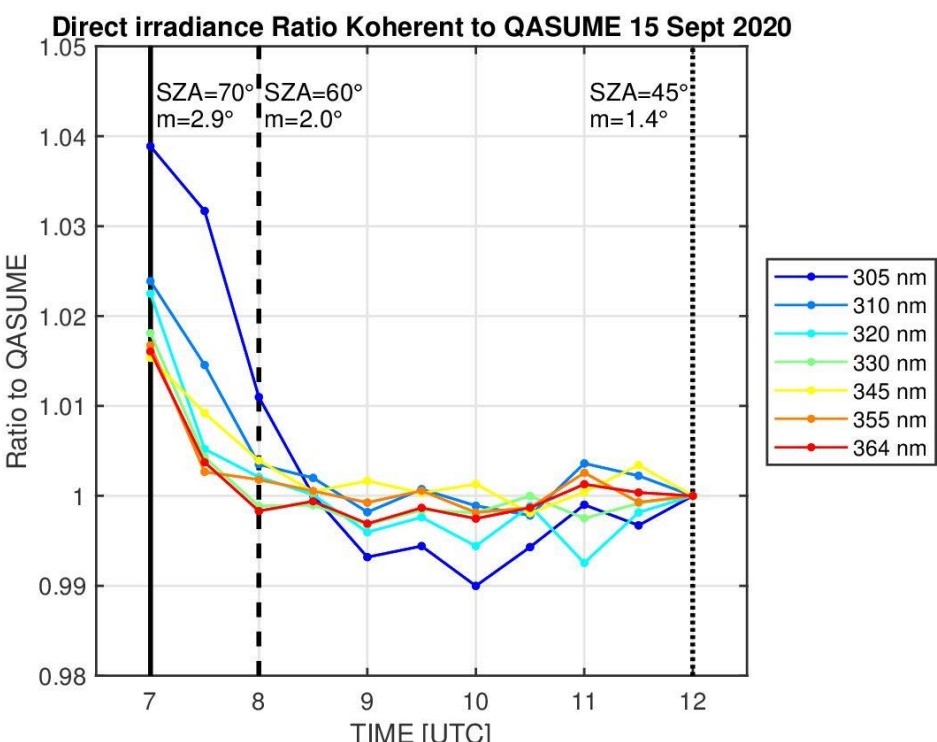


*Figure 2: Comparison between QASUME and Koherent spectra on the morning of 15 September 2020 displaying the direct irradiance ratios for individual wavelength bands within +/- 2.5 nm. The ratios are normalized to the ratio at 12 UTC (noon). The solar zenith angles (SZA) and the air mass (m) for UTC 7, 9, 12 (noon) are indicated by vertical lines.*


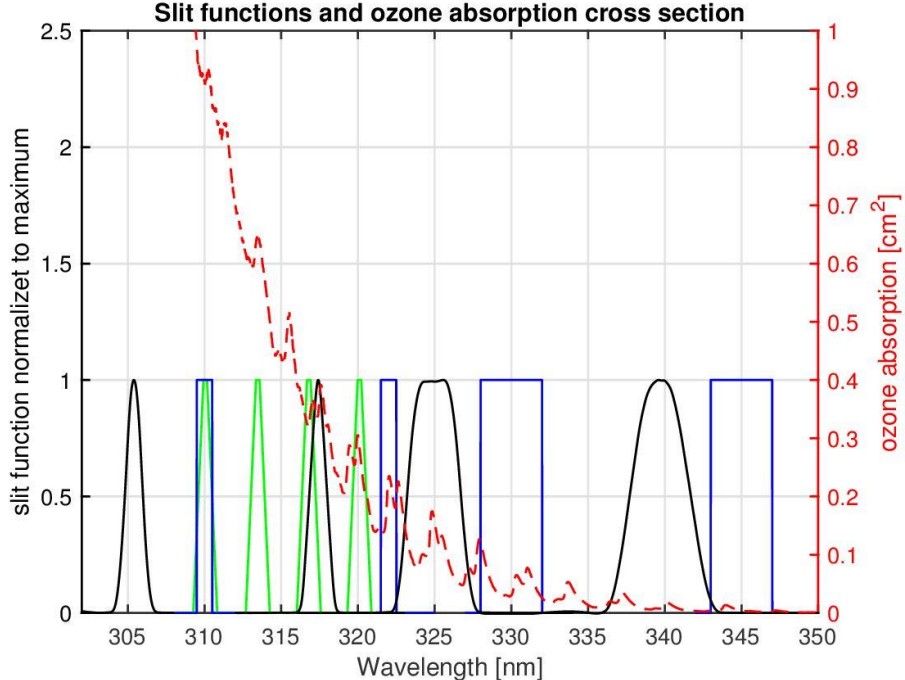

*Figure 3: Slit functions of the Custom Double Ratio technique for Koherent (blue), the trapezoidal slit functions of the Brewer (green) and the measured slit function of Dobson 101 (black). The red dashes line indicates the ozone absorption cross section SG14 at effective temperature of 228 K.*


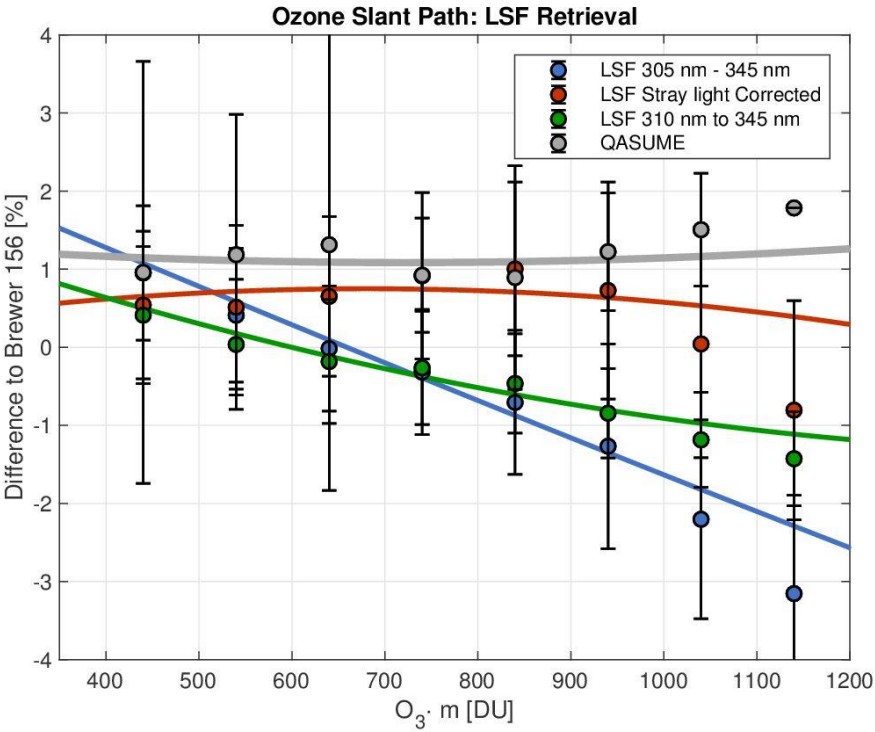

*Figure 4: Ozone slant path comparison of Koherent LSF retrieval with the double monochromator Brewer 156 for the wavelength range of 305 – 345 nm retrieval (blue), the wavelength range of 310 – 345 nm retrieval (green), and stray light corrected retrieval (red). For*
*comparison, the performance of QASUME is indicated in grey. The lines indicate quadratic fits to all measurements, which are not specifically shown. The circles are averages on 100 DU bins.*

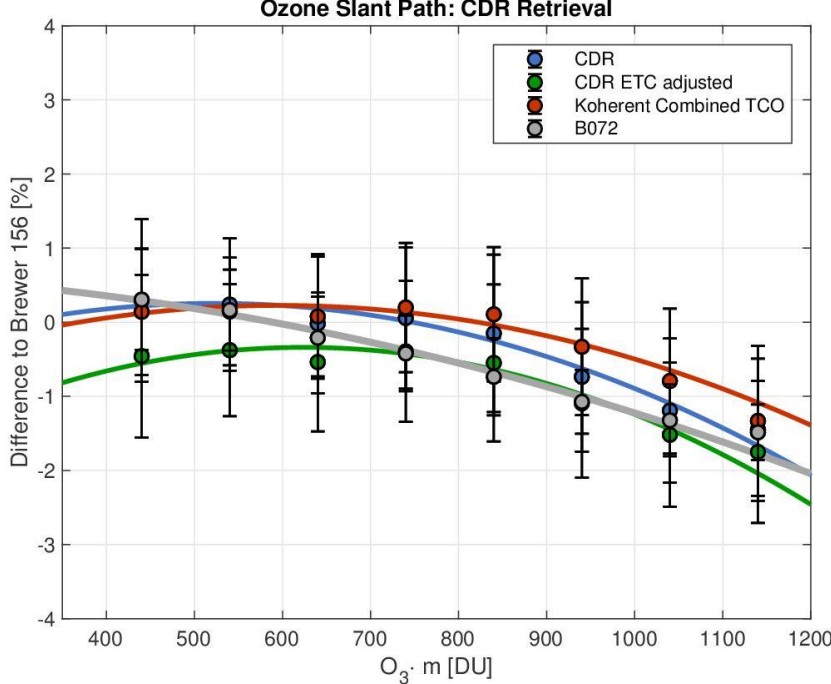

Figure 5: Ozone slant path comparison of Koherent CDR retrieval with the double monochromator Brewer 156 for standard CDR retrieval (blue), ETC adjusted CDR retrieval (green), and LSF/CDR combined retrieval (red). For comparison, the performance of the single monochromator Brewer 072 is indicated in grey. The lines indicate quadratic fits to all measurements, which are not specifically shown. The circles are averages on 100 DU bins.

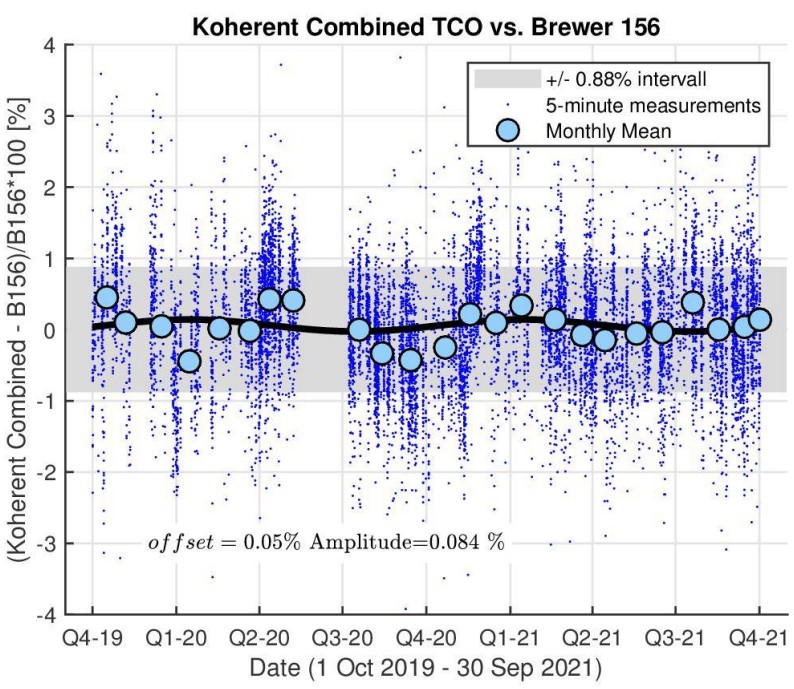

Figure 6: Relative difference between Koherent (LSF and CDR combined) and Brewer 156 double monochromator. The small blue points indicate the synchronized measurements in five minute interval. The light blue circles are monthly means. The grey area indicates the standard deviation of the relative difference between the three retrievals used for the combined product. The seasonal amplitude is 0.084% with a long term offset of 0.05%.

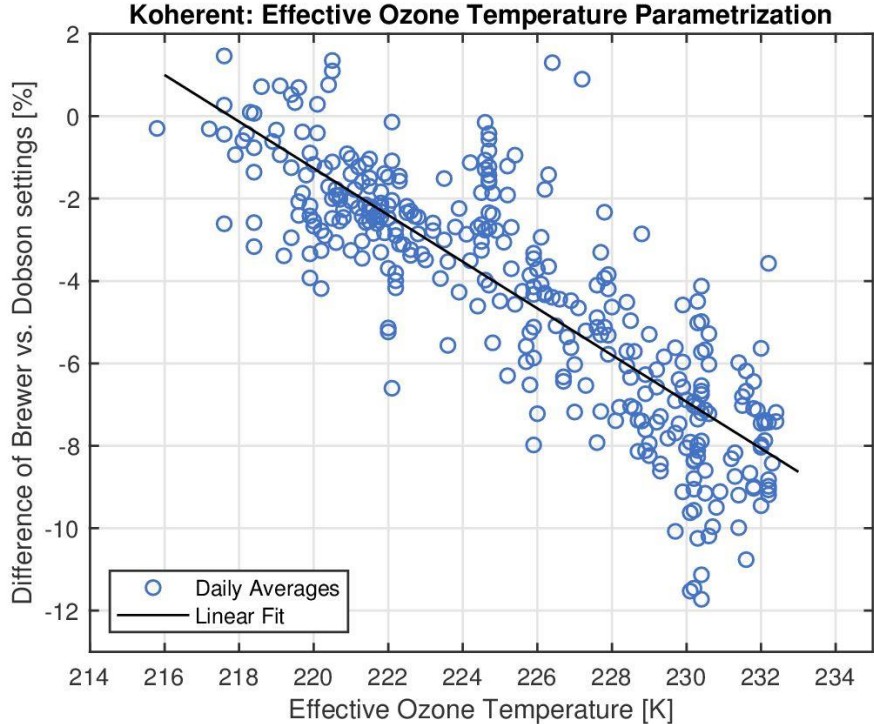


Figure 7: *Parametrization of the relative differences between CDR retrieval using Dobson and Brewer wavelength and slit function settings at 228 K effective ozone temperature, with daily effective ozone temperature from balloon soundings. The linear fit exhibits a slope of 0.5%/K indicating the sensitivity of the differences to effective ozone temperature.*


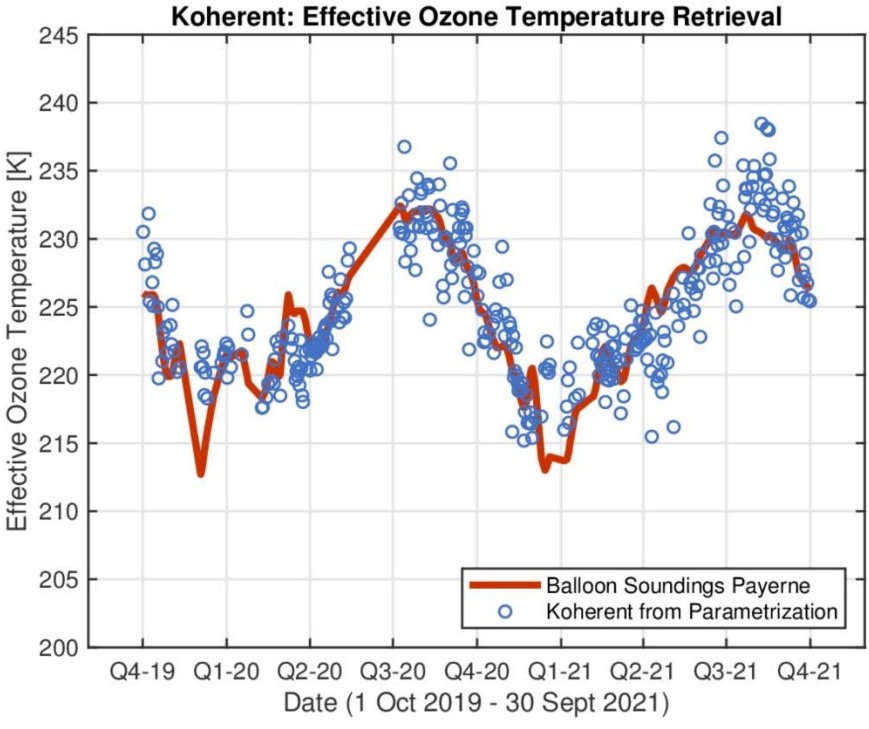


Figure 8: *Effective ozone temperature derived from the parametrization of differences between CDR retrieval from Brewer and Dobson wavelength and slit function settings.*