# Peer review of "Total Column Ozone Retrieval from a Novel Array Spectroradiometer"

_Atmospheric Measurement Techniques, 2022_

## Author Comment (AC1)

**Response to Reviewer #1**

Many thanks for the helpful work of Referee #1 on our manuscript. Below you can find our specific answers to the comments in blue. We have also revised the manuscript according to the suggestions by both Reviewers. The changes are marked with track changes in the word document. For extended changes we have indicated the line-number of the revised manuscript. We have acknowledged the two anonymous reviewers in the acknowledgements.

**General Comments**

The manuscript "Total Column Ozone Retrieval from Novel Array Spectroradiometer" published by Egli et al., presents a study on the use of a relatively new array spectroradiometer for ground-based measurements of total ozone column in the atmosphere. The new array spectroradiometer has the potential to provide more accurate and precise measurements compared to traditional methods such as those using Dobson instruments, grating spectrophotometers, etc. The study also provides a comparison of total column ozone retrievals between the proposed method and established methods. This helps to demonstrate the potential of the new system and the associated retrieval technique.

One of the strengths of this study is that it presents an approach for measuring total column ozone that is new, fast and automated, while utilizing available, easy to acquire software packages and hardware. The study also leverages on the established instrumentation and expertise at PMOD for standard calibrations.

However, I believe that this manuscript would still benefit from a chapter on error analysis containing a detailed error budget. I understand that some of the aspects of the methodology have already been done elsewhere, nevertheless it would be useful and important to include such a chapter. For example, I would be interested in knowing the signal to noise ratios of the spectra, typical wavelength shifts, how the uncertainty in the LSQ retrieval is calculated, and so on, without much digging through literature.

*We have included a paragraph with an overall uncertainty budget for both, the LSF retrieval and the CDR retrieval. With this paragraph we believe that the reader can follow how we have achieved that overall uncertainty budget. Lines 426 – 440 in the revised manuscript.*

In conclusion, this manuscript fits well within the scope of AMT. Therefore, I recommend its publication after addressing the general comments and some of the comments and corrections below.

*We thank the reviewer for this assessment of our publication.*

**Specific Comments**

The authors refer to the "low cost" of the Koherent system but do not provide any estimates of the costs involved and how they compare to available systems like the Brewer, BTS-Solar, etc.

*Koherent was first built as a research instrument and become later an operational instrument. The cost of the entire system was around 40'000 Euros. However, PMOD/WRC does not commercialise*

*this instrument. We refer to the commercial product from Gigahertz Optics GmbH, which provides the product "BTS-Solar". This instrument includes the same array spectroradiometer as in Koherent. We cannot state any prices from an external company, but the costs are a small fraction of a new Brewer.*

Can the authors please comment on any effects of UV radiation on the degradation of the optical fiber and if this would have noticeable effects over time?

*We agree that the degradation of the optical fiber could be a problem from a very long-term perspective. However, we have shown, that the system was stable over two years with only one calibration. We will further monitor the stability of the system to assess the long-term stability of the filters and the fiber. For operational use we recommend calibrating the system on a two years schedule. We have stated the two years stability in line 379 of the original manuscript.*

Why use an optical fiber instead of adopting a similar design to the BTS-Solar?

*We agree that this is a weak point of our system. Koherent was first built as a research instrument. The configuration with the optical fiber was chosen to test several additional filters with an in-line filter wheel and to test telescopes with different field of views. However, with the today's knowledge and for more simplicity, we would chose the more simple design with the collimation tube in front of the diffusor as it is provided by the commercial BTS-Solar. We have stated this in line 97 - 101 of the revised manuscript.*

BTS2048-UV-S-F array spectroradiometer: According to specifications, this spectroradiometer has a calibrated measurement range of 200 nm to 430 nm. Why do the authors truncate the upper wavelength range to 345 nm? Would it not be useful to include the maximum range covering the Fraunhofer lines at around 393 nm? I think this will make it easier to determine any wavelength shifts, will it not?

*Yes, for the wavelength shift a broader range of wavelength is better. For the wavelength shift we have selected the wavelength range from 295 nm to 370 nm. The range until 345 nm is for the ozone retrieval, because the ozone absorption is negligible above 345 nm (Figure 2).*

Two-point calibration: I don't quite understand the rationale of changing the absorption coefficients in addition to adjusting the ETC. It seems to me that the absorption coefficient is simply used as a "tuning parameter" in this case. Aren't the slit functions well determined, as well as the ozone cross sections? How would the authors explain the need to change the absorption coefficient?

*Indeed, that is a good point. In principle, Koherent does not require the adjustment of the absorption coefficient since this is calculated with the slit function and the ozone absorption cross section. In this paragraph, we wanted just to state that also a two-point calibration is possible, as it was used for Brewers in the past. We have removed this statement in the revised manuscript.*

The authors refer to minimal least squares, what do they mean by "minimal". A sentence or two explaining this would be sufficient.

*Minimal least squares means to minimize the sum of the squares of the offset to the fit (residuals). We agree that the word "minimal" is already included in "least". We have clarified this in the revised manuscript (line 188) and removed the word "minimal".*

**Technical Corrections and Suggestions:**

P.1, Line 24: "within less than 0.7%" --> within 0.7%

*Done*

P.2, Line 45: "In the Dobson instruments, prisms are selecting …" --> In the Dobson instruments, prisms are used to select …

*Done*

P2, Line 46: "Most of the Dobsons are manually operated and require therefore …" --> Most of the Dobsons are manually operated and therefore require …

*Done*

P2, Line 52: "contrary to single …" --> in contrast to single …

*Done*

P2, Line 52 and 69: "suffer from stray light …" --> suffer from the effects of stray light …

*Done*

P2, Line 53: "The Brewers were formed to a network of automatic stations, which required few …" --> The Brewers were used to form a network of automated stations, which required less …

*Done*

P2, Line 53: "best consistency …" --> greatest consistency

*Done*

P2, Line 60: "irradiance ratio at the top of the atmosphere" : irradiance ratio of what?

*Done*

P2, Line 76: "Similarly as Pandora …" --> Similar to the Pandora …

*Done*

P2, Line 83: "Contrary to the …" --> In contrast to the …

*Done*

P2, Line 83: "fiber coupled" --> fiber-coupled

*Done*

P9, Line 363: "clears sky" --> clear sky

*Done*

P9, Line 369: "The two-years …" --> The two-year …

*Done*

Fig. 1. Caption: "… spectra on morning of 15 September …" --> spectra on the morning of 15 September *Done*

---

## Author Comment (AC2)

**Response to Reviewer #2**

We acknowledge the detailed and helpful comments of Referee #2 on our manuscript. Below you can find our specific answers to the comments in blue. We have also revised the manuscript according to the suggestions by both Reviewers. The changes are marked with track changes in the revised word document. For extended changes we have indicated the line-number of the revised manuscript. We have acknowledged the two anonymous reviewers in the acknowledgements.

**General Comments:**

Egli et al. present the results of a two year comparison of total ozone measurements from a novel CCD array based instrument (named "Koherent") to a well-calibrated double monochromator Brewer (#156) at Davos.

Koherent is based on a modern array spectroradiometer manufactured by Gigahertz Optik, as used in the BTS-Solar instrument, but differing in that it is coupled by fibre to a telescope mounted on a solar tracker.

The instrument was previously described in Zuber at al. (2021) where a similar comparison to a Brewer in Davos was also reported, but here the retrieval technique has been refined and the comparison period extended.

Such work is extremely useful for the global ozone observing system, where there has long been a desire for a relatively cheap, automated and easy-to-maintain instrument to supplement or even replace (if the quality of measurements can be demonstrated to be high enough) the existing Dobson and Brewer networks. The subject of the manuscript is thus very suitable for AMT.

*We thank the reviewer for this favourable assessment.*

While the analysis presented is very sound, I believe the current version of the manuscript is deficient in terms of motivation and discussion of results, and needs enhancement before it is suitable for publication. Several options for the retrieval are presented and the results compared, but there is minimal discussion of motivation such as the theoretical advantages and disadvantages of the different methods (in particular, least squares fitting versus the ratio method) and how these are manifested in the results. In some places, the text states parameters have been optimized but there is no explanation given of how these optimizations were performed.

*We have now stated the motivation more clearly in the introduction lines 124 - 130. Generally, the advantage of the CDR technique is that this algorithm allows calibrating the instrument with dedicated ozone reference instruments such as the Brewer in periodic calibration field campaigns or with traceable to SI units instruments such as QASUME, by adjusting the extraterrestrial constant. Furthermore, this publication shows the advantage of deriving the effective ozone temperature from Koherent spectra analysis. The comments above are further addressed by the specific comments below.*

Rather than identifying or developing a preferred retrieval method, the authors made a decision to simply average the results of three different retrieval methods. This seems an odd choice to me and definitely needs to be properly explained.

*We agree that a merging of the different retrievals may not be applied for all applications. The reader can use the different retrievals regarding to its strength and weaknesses. However, since all retrievals are scientifically valid approaches all TCO could display the true ozone value. The average of all indicates how close the individual retrieval could be to the true value. This is indicated by the value of 0.88%. We have stated the motivation in the original manuscript in the introduction of section 3.3. For clarification we have added lines 415-417 in the revised manuscript.*

Also lacking from my perspective is the motivation of the design of the Koherent – what are its anticipated advantages compared to the BTS-Solar (particularly in the light of Zuber et al 2021 which overall seemed to favor the BTS-Solar configuration) or to the other CCD instruments mentioned in the introduction.

*Koherent was first built as a research instrument, to test several additional filters and telescopes and to apply a first version of the LSF retrieval method as presented in Zuber et al. 2021. LSF was not thoroughly investigated in Zuber et al. 2021. In particular the effect of stray light and effective ozone temperature was not well understood. To improve the data, we have addressed this in this publication (lines 103 – 107). We further developed and introduced the CDR technique to obtain a retrieval algorithm, which can be calibrated with standard in situ reference instruments (124– 130). In other words, we present here two well tested alternatives to the retrieval presented in Zuber et al. 2021. Unfortunately, this study could not compare the built in TCO retrieval algorithm from Zuber et. al. 2021 with the new LSF and the CDR technique, since BTS-Solar was not operated in Davos during this period. This may be subject to further research. However, we agree that the technical design of BTS-Solar with a collimating tube instead of a fiber-coupled telescope may be better in terms of simplicity (see also specific comments). We mentioned other instruments such as Pandora or Phaeton as other array spectroradiometer based systems. Unfortunately we cannot provide comparisons with these instruments in this publication. However, we plan to apply the double ratio technique also to Pandora spectra to test this alternative retrieval in comparison with the standard Pandora retrieval.*

I would also like to see at least a small comment about the anticipated long-term stability provided by the design of Koherent, in particular where degradation might be expected over time. This of course is a crucial aspect for an operational instrument.

*Since Reviewer #1 also commented this point, we repeat here the answer to reviewer#1: We agree that the degradation of the optical fiber and/or filters could be a problem from a very long-term perspective. However, we have shown, that the system was stable over two years with only one calibration. We will further monitor the stability of the system to assess the long-term stability of the filters and the fiber. For operational use we recommend calibrating the system on a two-year schedule. We have stated the two years stability in line 379 of the original manuscript.*

One unrelated general point is that the authors frequently use quite recent works (Gröbner et al. 2021 and to a lesser extent Redondas et al. 2014) as references for some longstanding points about Dobson and Brewers. I think it would be better scientific practice to cite the original literature in these cases.

*Thank you for this indication. We apologize for not citing the former literature adequately. We have added two new references in the revised manuscript (see section References and line 58-59).*

**Specific comments:**

Line 9 – I find this statement misleading because total ozone retrievals from Koherent have been presented before and compared to a co-located Brewer at Davos (ie in Zuber et al. 2021)

*We have re-worded the sentence: line 10 of the revised manuscript.*

Line 37 Please re-word this sentence for better clarity as it is not clear what exactly you are referring to – do you mean the date when ozone depletion was first identified in observations?

*We have re-worded the sentence: line 37-38 of the revised manuscript.*

Line 42 I suggest a reference to Figure 2 here would help the reader.

*The ozone absorption in the UV can be considered as general knowledge for an introduction. Since we would like to keep the introduction more general, mentioning figure 2 in the introduction may be a bit too much detail at this stage. Later in the manuscript, it will be shown in more detail and figure 2 will be explained in the text.*

Line 56-57 As I mentioned in the general comments, this is an example where I don't think it's appropriate to only mention the papers of Gröbner and Redondas.

*We have added three new references in line 58 (Kerr (2022), Vanicek (2066), Scarnato et al., (2009)) in line 58-59.*

Line 94 You should explain what the motivation is for this configuration.

*Since Koherent was first built as a research instrument, the configuration with the optical fiber was chosen to test several additional filters with an in-line filter wheel and to test telescopes with different field of views. However, a direct connection of the telescope with the fiber revealed the best configuration. However, with the todays knowledge and for more simplicity, we would chose the more simple design with the collimation tube in front of the diffusor as it is provided by the commercial BTS-Solar. We have stated this on line 97 – 101.*

Lines 95-112 I find the discussion of the different LSF algorithms confusing – you mention a "first version", then an "improved, standardized and optimized" version a "standard version … which is "considered here as the standard retrieval". However, you then go on to modify this algorithm further, eg applying a stray light correction. Saying "the standard LSF retrieval algorithm" might also be interpreted as meaning this algorithm is in wide use outside of your institution which I don't think you mean to imply.

*We agree that this paragraph is a bit confusing. In particular the word "standard LSF retrieval" is removed in the entire manuscript. We have clarified this in the revised manuscript, lines: 115 - 118*

Line 123 You say here "selecting the best retrieval" but I don't think you ever did that?

*Indeed, during development, we have tested several retrievals (but not extensively), but we agree that the few different testing should not be mentioned here. We have removed this in the revised manuscript.*

Lines 130-132 Just as a question, would the temperature stabilization be sufficient for extreme environments, eg desert, tropical or polar?

*We have not tested Koherent in such harsh environment. The cooling system of Koherent is the same as for QASUME which was operated in La Reunion (Tropical) and Ny Alesund (Polar) stabilizing the instrument to 20°C. However, direct sun impact is critical for the cooling system when ambient temperature is also high. QASUME needs to be shaded during operation at high ambient temperature and intense sun (which was not a problem in Davos). Since Koherent is equipped with an optical fiber, the box can also be placed indoors or in shaded environment.*

Lines 155-156 There also appears to be a small dip in the middle of the morning for 305 and 310 nm?

*We have included this observation in the revised manuscript. Line 170.*

Line 172 You should explain why you are using a different solar spectrum now.

*Since we have used the QASUME LSF retrieval, the new settings includes the solar spectrum from TSIS. We have stated this in the revised manuscript (lines 187-188) and clarified earlier in the original manuscript.*

Line 173 What does "non-linear" mean here?

*We mean just "least square fit". We have clarified in the revised manuscript lines 189-190.*

Line 190 The use of the Payerne data needs more detail - I take this to mean the total ozone value can only be calculated after the next ozonesonde flight so that you can use the interpolated 'daily input'? How are you treating the portion of the ozone profile that is above the burst height of the balloons?

*Thank you for this clarification. We have interpolated the sondes, which are provided 2-3 time a week to daily values. The ozone and temperature above the sonde burst height are obtained by extending the measured ozone and temperature profiles with the standard ozone and temperatures taken from the US Standard Atmosphere temperature and ozone density values just below the sonde burst height which are normalised to the sonde. We have included this in the revised manuscript. Lines 206 – 210.*

Line 194 Why wouldn't you use the value of pressure at the actual time of the measurement?

*The pressure in Davos varies about +/- 7 hPa in Davos during the year. It was shown that the used QASUME LSF algorithm is insensitive to that pressure variation (about 0.014% in TCO). Therefore we have chosen a climatological value. Since Brewers or Dobsons do not use the actual pressure as input for the retrieval, we have also chosen a climatological pressure of 820 hPa. We have considered this in the revised manuscript. Lines 213 – 214.*

Lines 185-197 You don't explain what values you are using for the ozone airmass and the aerosol airmass. Are you using the latest ozone profile from Payerne for the ozone airmass?

*The airmass $m$, denoting the path length of radiation through the atmosphere and is calculated based on the geometry between the Earth, atmosphere and the sun for each time. The airmass for the ozone ($m^{O3}$), aerosol ($m^{AOD}$) and Rayleigh ($m^R$) and SO2 ($m_\lambda^{SO2}$) is calculated from the*

*standard US atmosphere profile for mid-latitudes afglus (NOAA, 1976) as applied in Egli et al. (2022). We have stated this in lines 218- 221 in the revised manuscript.*

Commented [MBE1]: ??

Line 200 At this point, I would have expected to see some motivation for the double-ratio technique. What are its advantages and disadvantages compared to least-squares?

*We have added more explanation for the motivation in lines 124 - 126 and 226- 228.*

Line 204 "According to Gröbner " – I would prefer you to say "Following the notation of Gröbner "

*We have changed the manuscript accordingly. Line 230.*

Line 211 You should explain the reason why you "selected" these particular wavelengths.

*The selection of the wavelength is rather empirical and is based on a shift of 5 nm from the nominal wavelengths of the Dobson ($\lambda_i^{Dobson\ 101}$ =[305.4, 324.9, 317.4, 339.7] ). These wavelengths revealed a good performance, as other wavelengths within these wavelength ranges. However, all possible combinations of wavelength setting were not tested. We have stated this in lines 238 - 241 of the revised manuscript.*

Lines 221-223 Yes, but you should say why

*See above. The width of the slits are chosen in analogy to the Dobson slit width. For simplicity we have chosen a rectangular slit.*

Line 232 This makes it sound as if Gröbner et al. invented the double ratio technique!

*We have removed the citation here.*

Line 241 Explain "consistency with units" more clearly.

*Since the absorption cross section is measured in cm-2 TCO in Eq. 5 was multiplied by the factor 1000 to obtain DU. We have clarified this in the revised manuscript line 270.*

Line 243 But Section 2.2 also doesn't give any detail about how you are using the balloon soundings to calculate a daily effective ozone temperature

*See above and we have clarified in lines 206-210 of the revised manuscript.*

Line 248 As for the LSF, I am surprised you use the climatological pressure?

*Since Brewers and Dobsons do not require pressure, we have implicitly also used a climatological value for the CDR. A sensitivity analysis now showed that TCO varies to 0.09% at the variation of +/- 7 hPa in Davos. We have included this sensitivity in the overall uncertainty budget of the CDR retrieval (lines 279 and 426 – 440 of the revised manuscript). For future operational use of Koherent we will consider your suggestion to use actual station pressure. Please note that for other TCO retrievals (e.g. BTS-Solar) air pressure plays a more important role than when using LSF or CDR.*

Line 260 You say they are all "customized for Koherent" but you don't give any information about how this customization was undertaken.

*See above and we have clarified in lines 238-241 of the revised manuscript.*

Line 303 According to figure 3, there is only a small improvement from restricting the wavelength range, but in figure 1, the stray-light effect is quite large for 305 nm compared to the longer wavelengths – could you please comment on whether this is consistent?

*When using 305 nm to 345 nm the effect of stray-light is about 4.4% (Minimum to maximum). When using 310 nm to 345 nm the effect of stray-light is about 2.2% (Minimum to maximum). We consider this as a large improvement. The 2% difference may be remaining straylight also at 310 nm.*

Lines 313-315 What is the physical basis for the stray light correction, considering the previous discussion?

*There is no physical basis for the stray light correction. This means that we did not correct the spectra for stray light directly. As stated in line 343 we have parametrized the slant path dependency between the LSF retrieved TCO of QASUME (Egli et al. 2022) and the LSF retrieved TCO of Koherent by a linear fit.*

Lines 319-323 The stray light performance is certainly greatly improved but the offset seems to have slightly increased?

*This is correct. Table 1 shows that the offset increased to about 0.5% compared to non-stray light corrected LSF.*

Lines 324 Does this mean that, in general, Koherent would not be recommended for use for ozone slant paths greater than 900 DU? How does this compare to a Dobson?

*Table 1 shows that Dobson D101 shows a slant path dependency of 1.4%, while the straylight corrected LSF shows 0.5% in the range of 400 to 1100 DU. From this perspective Koherent LSF stray light corrected performs even better than Dobson D101. However, a larger decrease of Koherent between 1100 and 1200 DU slant path can be observed. From this perspective one may consider only to use TCO from Koherent below 1100. However, this judgement depends on the application.*

Line 351 The point is though that the retrieval assumes the slits have been correctly adjusted (eg Köhler et al. 2018 which is in your references)

*We have added this statement in the revised manuscript. Lines 382-383.*

Line 358 This confuses me – isn't this just calculated for Koherent based on your definitions?

*That is correct. In principle, Koherent does not require the adjustment of the absorption coefficient, since this is calculated. In this paragraph, we wanted to state that also a two point calibration is possible, as it was used for Brewers in the past. Reviewer 1 also mentioned this issue. We agree with your concerns and have removed this statement.*

Line 379 Have you actually shown that it is stable for two years? Are you saying this because of Figure 5?

*Yes, we argue because of figure5. We have drawn the two-years comparison of all the retrievals as in Figure 5 and made a linear fit through all data points. We have not seen any substantial drift during the two years.*

Lines 383-386 This seems very odd to me. What is the rationale for merging the results of three different retrievals, which would all presumably have their own strengths and weaknesses? There is only one true value of total ozone at any one time. This point needs more explanation.

*See general comment above and lines 415-417 in the revised manuscript.*

Lines 392-393 Do you mean 0.88% is the uncertainty corresponding to the difference between the three methods?

*Yes. It reflects the standard deviation between the retrievals as stated in the original manuscript. We have also included a more detailed uncertainty budget for LSF and CDR retrievals (see lines 426 – 440).*

Line 403 I don't think Redondas et al. 2014 was the first to report this fact

*Yes. Kerr (2002) is the better citation for this.*

Line 426 – As mentioned previously, how are you accounting for the portion of the profile above the burst height of the balloon? This might be contributing to the difference with the reanalysis.

*The burst height of the balloons is taken into account as explained above and included in lines 206 – 210 in the revised manuscript.*

Lines 329-429 From what I understand of your approach, the retrieval of temperature is almost accidental, in that you have applied a Dobson-style retrieval and a Brewer-style retrieval, and knowing the Dobson-Brewer difference is sensitive to effective ozone temperature, then gone on and calculated your regression. Couldn't this method be improved though, if you went back a step and selected from the start, a combinations of wavelengths chosen specifically for the retrieval of effective ozone temperature? Could you comment on this please.

*We have chosen the Brewer and Dobson style, since it is known that the Brewers are not sensitive to effective ozone temperature (Kerr, 2002), while the Dobsons are sensitive. From this perspective the approach is scientifically based. Maybe even better wavelength setting could be found. This may be subject for further research.*

Line 343 I agree this is a very useful advantage for operational instruments.

*We agree.*

Line 438 I don't this statement is quite true because this was already shown in Zuber et al. 2021.

*Please note that in Zuber et al. 2021 the comparison with Brewer 163 was using the Bass and Paur cross section (see lines 301 and 322). Here we have used the SG14 cross-section also for the Brewer. We have further clarified this in the revised manuscript. Lines 300 – 303 and 322.*

Line 459-461 I hope you will continue to monitor and assess the performance of Koherent over a longer time period and report on it.

*Yes, of course we will continue to monitor and assess the performance. Koherent is now part of the TCO observing instrument park at PMOD/WRC.*

Line 640  I find this plot slightly misleading, because the blue dots, the light blue circles and the black line all relate to the difference between Koherent and the Brewer, but the grey band relates to a different quantity, namely the difference between the three retrieval methods. I know this fact is stated in the caption, but I wonder if there is some way you could make this more intuitive visually for the reader?

*We agree that with the grey line a different comparison is shown. But we believe that showing the comparison with established instruments such as Brewer or Dobsons are important. The grey line should indicate that this is a different comparison and we believe that due to this light grey the line is modestly enough highlighted.*

**Minor comments**

Line 1 "From a novel array …"

*Done*

Line 52 Dobson-> Dobsons

*Done*

53-54 Please re-word this sentence for better grammar

*Done*

55 "are selecting" -> "prisms are used to select"

*Done*

Line 209 Were -> Where

*Done*

Line 254 "Analagous" -> "Analagously,"

*Done*

Line 273 "attributed to" -> "due to"

*Done*

Line 349 "good" -> "well" or "the performance is at least as good"

*Done*

---

## Author Response (AR2)

**Response to Reviewer (minor revision)**

Many thanks for your additional work on our manuscript. We appreciate your efforts. Below you can find our answer to the comments in blue. We have also revised the manuscript to clarify this open point. The changes are marked with track changes based on the first revised word document.

**General Comments**

I would like to thank the authors for their genuine engagement with the comments from both referees, and for the acknowledgement on lines 513-514.

Apart from some minor corrections listed below, there is only one response which I think has been misunderstood and where I would request some additional revision.

Regarding the ozone and aerosol airmass factor (originally lines 185-197 in the submitted manuscript, responded to in lines 218-221 in the revised manuscript), the authors state they have followed the US Standard Atmosphere from 1976. However, after many years of careful observation at Davos, I am sure better information than this must exist for the true vertical distribution of both ozone and aerosol at this location. Many previous works have considered the appropriate aerosol vertical distribution for measuring AOD in the UV, including some written by PMOD authors. I think the authors should state explicitly what height and distribution they are assuming for both aerosol and ozone, and comment on whether this would represent a significant source of uncertainty or not, particularly at higher zenith angles. The Payerne ozonesonde record would provide a climatology of effective ozone height.

*We agree that more information about the vertical distribution profile could improve the retrieval of TCO. We have addressed the impact of the difference between the standard atmosphere and the ozone sondes from Payerne on TCO retrieval for a) double ratio technique in Gröbner et al. (2021) and for the LSF retrieval in Egli et al. (2022). In summary, the averaged effective ozone height over Payerne is 22.3 km with a minimum of 20.2 km (winter) and a maximum of 23.8 km (summer). The difference of 3.6 km results in a maximum of 0.3% difference in TCO at air mass of 3.9 (SZA 76), which is the maximum that can be reached due to the mountainous horizon of Davos. For comparability with Brewer 156 we have chosen a constant ozone layer of 22 km and the effective molecular scattering height relevant for Rayleigh scattering of 5 km as discussed in Gröbner et al. (2021).*

*The effect of the variation of the effective ozone height is implicitly considered in this publication in the uncertainty budget for both the LSF retrieval and the CDR retrieval (lines 425 – 439). For the LSF retrieval, the air mass uncertainty is included in the statement that the contribution of the other uncertainties remains the same as in Egli et al. (2022). Regarding the CDR retrieval, the uncertainty from air mass of 0.085% (k=1) is explicitly stated in line 432. Since 0.3% are maximum values we have derived the k=1 (1 sigma) uncertainty by 0.086%=0.3%/2/sqrt(3) as discussed in Egli et al. (2022).*

*We agree with the reviewer that we should explicitly state the vertical height for ozone (22 km) and aerosol (5 km) layer we have used. We have clarified this in the description of the CDR retrieval lines 258-263 and included in 439 (uncertainty budget of CDR). For further operational long-term measurements and to further reduce the uncertainty we consider including the individual soundings from Payerne and local ground pressure. We thank the reviewer for this suggestion.*

**Minor Comments**

Line 18 Has it really been 'optimized'?

*Done. We have removed "optimized"*

Line 26 "as good" -> "as well"

*Done.*

Line 30 "similar as" -> "similar to"

*Done.*

Line 32 "allows to determine" -> "allows the determination of"

*Done.*

Line 33 Insert space between "to" and "monitor"

*Done.*

Lines 33-34 "The presented TCO retrieval" -> "The TCO retrieval presented" or "The TCO retireval presented here"

*Done.*

Line 37 "is" -> "was"

*Done.*

Line 38 "and since then" -> "and since then has been"

*Done.*

Line 54 Insert space between "network" and "still"

*Done.*

Line 61 I would prefer to say the ETC defines the inferred response of the instrument to the irradiance at the top of the atmosphere

*Done.*

Line 104 "is attributed to" -> either "should be attributed to" or "is caused by"

*Done.*

Line 123 "allows to retrieve TCO" -> "allows TCO to be retrieved" or "allows retrieval of TCO"

*Done.*

Lines 123-124 " the measured spectra of Koherent are used to apply" doesn't make sense , I suggest something like "the measured spectra of Koherent are used as an input" or "a custom double ratio technique can be applied to the measured spectra".

*Done.*

Line 237 Insert space between "than" and "other"

*Done.*

Line 286 Again, I think "customized" is misleading, given this doesn't seem to have been a particular aim of your work

*Done. We use "chosen" instead of customized.*

Line 437 "to about" -> "to be about"

*Done.*

Line 487 "method" -> "methods"

*Done.*

Line 487 "such as" -> "namely" or another similar word

*Done.*

*We have further removed other typos in the revised manuscript.*